# Epigenetic and Drug Response Modulation of Epigalocaten-In-3-Gallate in *Staphylococcus aureus* with Divergent Resistance Phenotypes

**DOI:** 10.3390/antibiotics12030519

**Published:** 2023-03-04

**Authors:** Ana Rita Mira, Ana Sofia Zeferino, Raquel Inácio, Mariana Delgadinho, Miguel Brito, Cecília R. C. Calado, Edna Ribeiro

**Affiliations:** 1H&TRC—Health & Technology Research Center, ESTeSL—Escola Superior de Tecnologia da Saúde, Instituto Politécnico de Lisboa, Av. D. João II, lote 4.69.01, Parque das Nações, 1990-096 Lisboa, Portugal; 2Hospital do Espírito Santo de Évora, E.P.E., Largo do Sr. da Pobreza, 7000-811 Évora, Portugal; 3Escola Superior de Tecnologia da Saúde, Instituto Politécnico de Lisboa, Av. D. João II, lote 4.69.01, Parque das Nações, 1990-096 Lisboa, Portugal; 4Centro Hospitalar de Lisboa Central, Hospital Curry Cabral, Rua Beneficência, 8, 1050-099 Lisboa, Portugal; 5CIMOSM—Centro de Investigação em Modelação e Otimização de Sistemas Multifuncionais, ISEL—Instituto Superior de Engenharia de Lisboa, Instituto, Politécnico de Lisboa, R. Conselheiro Emidio Navarro 1, 1959-007 Lisboa, Portugal

**Keywords:** healthcare-associated *S. aureus* infections, epigallocatechin-3-gallate (EGCG), synergistic effects, drug resistance phenotypes, epigenetic modulator, drug resistance modulators

## Abstract

Healthcare-associated methicillin-resistant *Staphylococcus aureus* infections represent extremely high morbidity and mortality rates worldwide. We aimed to assess the antimicrobial potential and synergistic effect between Epigalocatenin-3-gallate (EGCG) and different antibiotics in *S. aureus* strains with divergent resistance phenotypes. EGCG exposure effects in epigenetic and drug resistance key modulators were also evaluated. *S. aureus* strains (*n* = 32) were isolated from infected patients in a Lisbon hospital. The identification of the *S. aureus* resistance phenotype was performed through automatized methods. The antibiotic synergistic assay was performed through disk diffusion according to EUCAST guidelines with co-exposure to EGCG (250, 100, 50 and 25 µg/mL). The bacteria’s molecular profile was assessed through FTIR spectroscopy. The transcriptional expression of *OrfX*, *SpdC* and *WalKR* was performed by using qRT-PCR. FTIR-spectroscopy analysis enabled the clear discrimination of MRSA/MSSA strains and the EGCG exposure effect in the bacteria’s molecular profiles. Divergent resistant phenotypes were associated with divergent transcriptional expression of the epigenetic modulator *OrfX*, particularly in MRSA strains, as well as the key drug response modulators *SpdC* and *WalKR*. These results clearly demonstrate that EGCG exposure alters the expression patterns of key epigenetic and drug response genes with associated divergent-resistant profiles, which supports its potential for antimicrobial treatment and/or therapeutic adjuvant against antibiotic-resistant microorganisms.

## 1. Introduction

The escalation of antibiotic-resistant human pathogens [1] is currently characterized by the World Health Organization (WHO) as a worldwide health challenge [2]. This reality has created an urge to develop new therapeutic drugs and molecular targets with therapeutic potential.

In *Staphylococcus*, the main targets of antibiotics are the cell envelope, ribosome and nucleic acid biogenesis or metabolism [3]. However, *S. aureus* acquired an insertion in the mecA gene via horizontal genetic transfer, the *Staphylococcal Cassette Chromosome mec complex* (*SCCmec*), which encodes a modified penicillin-binding protein 2 (PBP2), is accountable for the phenotype resistant to antibiotics (oxacillin, streptomycin, tetracycline, erythromycin, among others), due to the decrease in the affinity of antibiotic–receptor binding, originating the *methicillin-resistant S. aureus* (MRSA) [4,5].

MRSA is currently acknowledged as one of the most relevant human pathogens, particularly in clinical settings, due to the fact that it is a major cause of infections worldwide [6] and is associated with extremely high mortality rates for invasive bloodstream and pneumonic infections [7]. Healthcare-associated MRSA infections (HA-MRSA) are typically associated with invasive procedures or devices and with divergent molecular profiles [8]. Relevantly, the ability demonstrated by this particular pathogen to develop new clones with the capacity to invade community settings and infect people without any known predisposing risk factors has become an issue of great concern [4].

The reversal of this resistance phenotype has been reported by exposure to Epigallocatechin-3-gallate (EGCG), a natural component and a major constituent of green tea [9]. EGCG has a wide range of health benefits, including its use as an antimicrobial agent against Gram-negative and Gram-positive bacteria (e.g., *E. coli*, *Salmonella* spp., *S. aureus* and *Enterococcus* spp.), some fungi (e.g., *Candida albicans*) and viruses (e.g., HIV, Herpes simplex and Influenza) [9]. Moreover, with its co-addition with antibiotics, such as β-lactam antibiotics, including penicillin, ampicillin and even carbapenems, macrolide tetracycline and t aminoglycosides gentamicin, may reduce the antibiotic’s minimum inhibitory concentration (MIC) and, consequently, increase MRSA’s susceptibility to that antibiotic [10,11,12,13].

Human exposure to EGCG affects a diverse range of metabolites in human plasma [14]. Previous studies classified EGCG peak serum concentrations in the range of 100 to 400 ng/mL as safe in human clinical trials [15]. Relevantly, EGCG-associated effects have been associated with this catechin potential as an epigenetic modulator as it is able to affect DNA methylation and gene expression patterns through the inhibition of DNA methyltransferases (DNMTs) or the activity of Histone deacetylases (HDACs) [16]. In *Staphylococcus*, one of the most relevant epigenetic modulators is *orfX*, which encodes for a 70S ribosomal methyltransferase whose substrate is S-adenosyl-L-methionine, and its C ‘terminal is inserted into the *SCCmec* complex, which contains the *mecA* gene [17].

Furthermore, antibiotic resistance is also mediated by the expression of key drug resistance genes, such as *WalKR*, a two-component system known to be crucial for the rapid adaptation of *S. aureus* to a wide range of environmental conditions through the production of autolysins [18,19], which controls several relevant staphylococcal virulence genes and triggers host inflammatory responses [19]. The WalKR system positively regulates the expression of the *spdC* gene and the *SpdC* protein, a newly described virulence factor, which in turn, decreases the expression of genes in the WalKR system. Interestingly, mutants for the *spdC* gene showed a decreased capacity for biofilm formation and a reduced expression of multiple virulence genes [20].

Considering the fact that epigenetic and drug resistance modulators are currently seen as potential targets for new therapeutic approaches for human pathogens, this work aims to evaluate the potential of EGCG in reversing the resistant *S. aureus* phenotype in strains isolated from hospital nosocomial infections and to assess its effect on the transcription of target epigenetic and drug resistance modulator genes. This study aims to point out the EGCG potential for antimicrobial treatment and/or therapeutic adjuvant against resistant microorganisms and epigenetic modulator effects associated with divergent resistance profiles.

## 2. Results

### 2.1. Exposure to EGCG Induced a Synergistic Effect with Amoxicillin in MSSA Resistant Phenotypes

Potential synergism between EGCG and the different antibiotics tested was assessed in the hospital strains identified as resistant after the inoculation of different concentrations of EGCG (250, 100, 50 and 25 µg/mL). Regarding the results obtained for the evaluation of potential synergism with different concentrations of EGCG with amoxicillin 25 µg, we observed that the strains coded as MS2 and MS12 (MSSA) presented with a resistance phenotype, but when exposed to EGCG in concentrations of 250 µg/mL, 100 µg/mL and 50 µg/mL, the phenotype reverted to being sensitive. The strains encoded as MS1, MS3, MS4, MS8, MS9, MS11, MS14, MS15 and MS16 maintained the resistance phenotype; that is, no synergism was observed between EGCG and amoxicillin in these strains. Regarding the MRSA strains, all of the strains maintained the resistance phenotype when exposed to EGCG at different concentrations.

### 2.2. EGCG Reverted the Resistant Phenotype to Gentamicin in MSSA and MRSA Strains

The evaluation of the potential synergism of EGCG with gentamicin 30 µg in MSSA strains demonstrated that for the MS13 strain (MSSA), EGCG exposure had no synergistic effect with gentamicin (30 µg). All of the other isolated MSSA strains presented with a phenotype that was sensitive to gentamicin resistance (Table 1). On the other hand, in MRSA strains encoded as MR3 and MR4, which initially exhibited a gentamicin resistance phenotype, after exposure to all of the tested concentrations of EGCG, the phenotype was completely reversed. The strain encoded as MR15 exhibited a reversion to gentamicin resistance after exposure to 250 µg/mL and 100 µg/mL EGCG concentrations, whereas, in the case of 50 µg/mL and 25 µg/mL of EGCG, the phenotype remained resistant (Table 1).

### 2.3. No Alterations in Imipenem Resistance Were Observed Associated with EGCG Exposure

All of the isolated MSSA strains had a sensitive phenotype regarding imipenem. On the other hand, the MRSA strains coded as MR3, MR7, MR11 and MR12 with associated resistance maintained the resistance phenotype after exposure to the different assayed concentrations of EGCG (data not shown). 

### 2.4. EGCG Affects Zone of Inhibition Values after Co-Exposure with Imipenem, Tetracycline, Gentamycin and Amoxicillin

Regarding time exposure effects in EGCG co-exposure with the tested antibiotics, we only observed a significant divergence in MSSA amoxicillin between 24 h and 48 h of exposure (*p* = 0.010). Furthermore, at the same exposure time, we also reported differences between MSSA and MRSA strains, namely after EGCG co-exposure with imipenem and amoxicillin for 18 h (*p* = 2.4 × 10^−5^ and *p* = 6.4 × 10^−8^, respectively), 24 h (*p* = 3.2 × 10^−5^ and 4.1 × 10^−9^, respectively) and 48 h exposures (*p* = 4.8 × 10^−5^ and 1.2 × 10^−8^, respectively). Moreover, when we analyzed the divergences between the data from commensal and nosocomial isolated strains, all of the exposure times and tested antibiotics resulted in significant differences (*p* ≤ 0.001), except for imipenem co-exposure with EGCG in MSSA strains, with no significant alterations observed after 18 h, 24 h and 48 h.

### 2.5. Exposure to EGCG Alters S. aureus Molecular Profile 

The strains selected for molecular profile analysis based on FTIR spectroscopy presented divergent resistance phenotypes after EGCG exposure (Table 2). 

FTIR spectroscopy reflects the vibrations between atoms within molecules according to the atoms and bonds associated and from the interactions due to the atoms surrounding them in a highly sensitive and specific mode. Consequently, the technique captures the bacteria’s whole molecular fingerprint, which can be used to discriminate between bacteria at the genus, species and clonal levels [22,23,24]. According to this, in the present work, the technique was applied to evaluate the discrimination between MRSA and MSSA strains (Figure 1). It was observed that the spectra of MRSA and MSSA present distinct molecular features, as demonstrated in the spectra principal component analysis and highlighted by the 100% separation between MSSA and MRSA in a spectra hierarchical cluster analysis (Figure 1E). It is worth pointing out that non-derivatized and derivatized spectra were always normalized to minimize the impact of biomass quantity in the molecular profile retrieved by the spectra. Derivatized spectra may resolve bands, increasing the information that can be extracted. However, derivatives also increase noise, and therefore, the non-derivatized and derivative spectra were analyzed. 

Due to the high sensitivity and specificity of FTIR spectroscopy when capturing the bacteria’s molecular fingerprint, the technique has also been used to evaluate the impact of drugs on the bacteria’s metabolism [25,26,27]. According to this, in the present work, the technique was applied to evaluate the impact of EGCG on the MSSA and MRSA strains. As can be observed in Figure 2B, all of the MSSA strains not incubated with EGCG (blue scores) are in a different region of the score- lot after EGCG exposure (red scores), and consequently, all MSSA strains presented a different molecular composition after EGCG exposure. In Figure 2A, all of the MRSA strains not incubated with EGCG (green scores) are also on a different region of the score plot in relation to the same strain after EGCG exposure (grey, brown and orange colors). Therefore, all of the MRSA strains presented a different molecular composition after EGCG exposure. Hence, the bacteria’s whole molecular composition was affected by the EGCG exposure, independently of the strains and over all concentrations of EGCG evaluated. 

Another interesting observation from Figure 2A is that the distance between the scores of strains after EGCG exposure is higher with MRSA than with the MSSA strains. That is, the distances in the score plot of the MRSA strains with or without 100 mg/mL EGCG (green versus orange, Figure 2A) was higher than the distance between MSSA with or without 100 mg/mL EGCG (blue versus red, Figure 2A). This points to the fact that EGCG has a higher impact on the MRSA strains than on the MSSA strains. Furthermore, the incubation of the MRSA strains with EGCG apparently resulted in strains with a lower diversity of molecular composition in relation to the strains not exposed to EGCG, as the samples are less dispersed over the score plot (Figure 2A,B). Based on the spectra PCA, it was not possible to discover a pattern associated with the resistance reversion due to EGCG exposure. 

### 2.6. Exposure to EGCG Affects Transcriptional Patterns of Epigenetic and Drug Resistance Modulators Genes

For the transcriptional analysis of the epigenetic modulator (*OrfX*) and drug resistance (*SpdC* and *WalKR*), the gene expression strains were considered regarding the divergent resistance phenotypes obtained after EGCG exposure (Table 3). MS16 (for MSSA strains) and MR22 (for MRSA strains) were the most resistant strains and thus utilized for a transcriptional expression comparison regarding the other selected strains. 

#### 2.6.1. Transcriptional Effects after Co-Exposure with EGCG in Selected MSSA and MRSA Strains

The relative expression of the genes *Spdc*, *WalR* and *Orfx* was analyzed using qRT-PCR in the selected strains with divergent resistant phenotypes. For the transcriptional analysis of the selected MSSA strains (Figure 3A), the MS16 strain was considered to be the most resistant due to the fact that, similar to the other strains, it was sensitive to imipenem and tetracycline; however, it maintained resistance to gentamicin and amoxicillin after EGCG exposure, and thus no reversion was observed in the phenotype. MS3 was sensitive to imipenem, tetracycline and gentamicin but resistant to amoxicillin, and, after EGCG exposure, no reversion was observed in the phenotype and *Spdc* mRNA levels were upregulated (6.38 ± 0.11 *p* < 0.001). MS5 strains, which exhibited a reversion of the resistant phenotype for *amoxicillin* up to 50 µg/mL of EGCG and *Orfx* mRNA levels, were upregulated (2.87 ± 0.15 *p* < 0.001). The MS13 strain was sensitive to all of the assessed antibiotics, and all of the analyzed genes presented with an increased transcriptional expression, namely *Spdc*, *WalR* and *Orfx* (14.37 ± 0.83; 2.06 ± 0.06; 1.3 ± 0.06; *p* < 0.001, respectively). Conversely, the MS15 strain was sensitive to imipenem, tetracycline and gentamicin but resistant to amoxicillin, and, after EGCG exposure, no reversion was observed in the phenotype, the *Spdc* mRNA levels were upregulated (4.25 ± 0.007; *p* < 0.001), but we observed a downregulation of *WalR* and *Orfx* transcriptional expression (−1.377 ± 0.12 *p* < 0.01; −2.16 ± 0.14; *p* < 0.001, respectively). Regarding the isolated MRSA and selected strains, the transcriptional analysis revealed a lower effect of EGCG exposure on the mRNA expression of *Spdc*, *WalR* and *Orfx* genes (Figure 3 B). The MR22 strains were considered to be the most resistant strains as they exhibited a phenotype sensitive to tetracycline and gentamicin; however, they maintained resistance to imipenem and amoxicillin after EGCG exposure. MR7 was sensitive to imipenem, tetracycline and gentamicin but resistant to amoxicillin, and, after EGCG exposure, no reversion was observed in the phenotype. The MR18 phenotype was sensitive to imipenem and tetracycline but resistant to gentamicin, which was reversed after EGCG exposure, and also maintained resistance to amoxicillin. MR31 was sensitive to imipenem and resistant to gentamicin and amoxicillin, in which we reported a reversion of up to 100 ug/mL of EGCG for gentamicin. In all these strains, only Orfx mRNA levels were upregulated (MR22: 2.46 ± 0.098 *p* < 0.001; MR7: 1.6 ± 0.22 *p* < 0.01; MR31 2.97 ± 0.35 *p* < 0.001). In the MR9 strain transcriptional analysis, which was resistant to imipenem and amoxicillin, there was no observed reversion after EGCG exposure but a total reversion of the resistance to gentamicin; no effects were observed regarding the mRNA levels. 

#### 2.6.2. Transcriptional Effects after 24 h Subculture of MSSA and MRSA Strains with Previous Co-Exposure with EGCG

Gene expression analysis performed after a 24 h subculture of the selected strains with no EGCG exposure demonstrated how the expression patterns observed were generally maintained immediately after exposure, with some divergences (Figure 4). MS3 maintained the upregulation of *Spdc* mRNA levels (6.05 ± 0.16 *p* < 0.001). MS5 also maintained the upregulation of *Orfx* mRNA levels (3.6 ± 0.17 *p* < 0.01) and additionally increased levels of *SpdC* and *WalKR* transcription (2.2 ± 0.06, 0.4 ± 0.03 *p* < 0.01, respectively). Conversely, the MS13 strain maintained *Spdc* upregulation (13 ± 0.17; *p* < 0.001)*;* however, a downregulation of *WalR* and *Orfx* was observed (−3.6 ± 0.12; −3.8 ± 0.04; *p* < 0.001, respectively). Additionally, in the MS15 strain, *Spdc* mRNA levels were upregulated (7.7 ± 0.25; *p* < 0.001), as observed previously, in addition to *WalR* and *Orfx* transcriptional expression (2.38 ± 0.07 *p* < 0.001; 2.24 ± 0.29; *p* < 0.01, respectively). Regarding the selected MRSA strains, *Orfx* upregulation was maintained in all strains, namely MR7, MR9, MR18 and MR31 (0.48 ± 0.06 *p* < 0.01; 1.04 ± 0.13 *p* < 0.01; 4.8 ± 0.04 *p* < 0.001; 1.76 ± 0.12 *p* < 0.01, respectively). However, MR7 and MR31 also presented increased transcriptional levels of *Spdc* mRNA (0.89 ± 0.05; 1.3 ± 0.08; *p* < 0.01, respectively), and in MR18 strains, *SpdC* and *WalKR* transcription was also upregulated (4.9 ± 0.32; 4.73 ± 0.1; *p* < 0.001, respectively). 

## 3. Discussion

*Staphylococcus aureus* is a commonly isolated pathogen in both hospitals and the community. These microorganisms are hastily spread worldwide and encompass a wide variety of antimicrobial resistance (including MRSA), which has become a major threat to human health [28]. *Staphylococcus aureus* strains isolated from acute nosocomial infections enclose divergent resistant profiles to the antibiotics commonly utilized as therapeutic options, as we can confirm by the results obtained (Table 1). 

Nosocomial strains of MRSA are typically multi-resistant and are associated with SCCmec type I, II or III, which comprise genes that code for resistance to various classes of antibiotics [29,30,31,32,33]. These strains are usually treated with a combination of two oral antimicrobials [34]; however, the escalation of divergent resistant phenotypes has created a need to develop new therapeutic alternatives, preferentially with lower hazardous side effects. Currently, one of the new therapeutic approaches under clear development is the use of natural compounds, such as green tea catechin EGCG. This catechin has been associated with several beneficial health effects [9,35,36] and anti-infectious properties against both Gram-negative and Gram-positive bacteria, some fungi and viruses [9,37,38]. Additionally, EGCG has also been described as a relevant epigenetic modulator with the capacity to affect gene expression patterns through the inhibition of DNMTs or the activity of HDACs [16].

In this study, we have primarily characterized the phenotypic resistant profiles of MSSA and MRSA strains isolated from hospital infections in a microbiology laboratory from a hospital in Lisbon. We evaluated the synergistic potential of EGCG with commonly utilized antibiotics and its effects on epigenetic and drug-resistant modulator genes. Additionally, we have also assessed the FTIR spectra of *S. aureus* strains, reflecting the vibration modes of the high diversity of chemical functional groups [39] and, consequently, representing the bacteria’s whole molecular composition.

Our data enabled the clear discrimination of MRSA from MSSA strains. These results are in agreement with other authors’ observations that demonstrate the high sensitivity and specificity of FTIR-spectroscopy in retrieving the molecular cell fingerprint that can be used for *S. aureus* typing [40,41]. Moreover, we observed that the bacteria’s whole molecular profile was clearly altered after EGCG exposure for all analyzed concentrations for both MRSA and MSSA strains. However, we could not observe a profile associated with resistance reversion to EGCG exposure.

The data obtained demonstrated that after EGCG exposure, only two isolated MSSA strains reverted the resistant phenotype to amoxicillin at concentrations of 250 µg/mL, 100 µg/mL and 50 µg/mL of EGCG. Additionally, all of the MRSA strains maintained the resistant phenotype to this antibiotic; that is, no synergism with EGCG was observed. On the other hand, in the results obtained regarding EGCG’s interaction with gentamicin, for MSSA strains, only one was resistant, and after EGCG exposure, it maintained the resistant phenotype. On the other hand, for MRSA, in the three strains with initially resistant phenotypes, when exposed to EGCG, two reverted to being sensitive, and one only reversed the phenotype to being sensitive after exposure to EGCG concentrations of 250 µg/mL and 100 µg/mL. Gentamicin is a broad-spectrum antibiotic in the aminoglycoside class [12], which acts by binding to the 30S ribosomal subunit, incapacitating the bacteria from synthesizing proteins [3,12]. Previous studies have reported that the combination of antibiotics, such as tetracycline, gentamicin, penicillin, β-lactams and carbapenems, with EGCG, has been shown to reduce the minimum inhibitory concentration (MIC) of antibiotics and increase the susceptibility of MRSA to these, revealing a high synergistic potential [10,11,12,13,37,42,43]. Previous studies have provided more insights into the effects of EGCG and have shown that concentrations of 50 mg/mL are capable of reversing tetracycline resistance in *Staphylococcus* isolates [44,45]. Additionally, green tea extracts were reported to reverse the resistant phenotype in MRSA strains [46], and EGCG at 25 µg mL concentrations was able to change the resistant phenotypes [43].

Moreover, our data also suggested that the length of exposure is relevant, as we observed significant differences in the MSSA strains exposed to amoxicillin and EGCG between 24 h and 48 h. Additionally, EGCG co-exposure to imipenem and amoxicillin resulted in clear divergences in MSSA and MRSA strains after 18 h and 48 h of exposure. 

During the progression of *S. aureus* from an asymptomatic colonizer in the community to an invasive pathogen associated with nosocomial infections, such as the strains tested in the present study, this microorganism acquires differential virulence factors, such as adhesin genes, immune evasion genes and toxins [47]. Nosocomial strains typically show multidrug resistance, representing a challenge for existing therapeutics. Thus, assessing the effects of new compounds on resistance modulator genes is crucial to sustaining their potential for new therapeutic approaches. 

Here, we assessed the effects of EGCG exposure on *WalR* transcription levels, a two-component system essential for the viability of *S. aureus* that actively participates in cell wall metabolism [18,19]. *WalR* positively regulates autolysis, biofilm production and alpha-hemolytic activity [19,20]. It also positively regulates major virulence genes, such as the *SpdC* gene [19,20]. On the other hand, the constitutive activation of *WalR* strongly diminishes *S. aureus* virulence, inducing an early triggering of the host’s inflammatory response, including neutrophil recruitment and increased cytokine levels [19]. These processes result in rapid bacterial clearance and lowered virulence [19]. In the present study, EGCG did not affect *WalR* transcription patterns, suggesting that the strains tested have high levels of virulence and interfere with resistance phenotype reversion since none showed a total reversion of the amoxicillin-resistant phenotype after EGCG exposure. 

Furthermore, we also assessed the effects of EGCG on *SpdC* gene transcription levels. The *SpdC* gene product negatively regulated the *WalKR* system and increased *SpdC* transcription levels, which are associated with higher virulence and diminished sensitivity to β-lactam antibiotics [20]. After a 24 h subculture of MSSA and MRSA strains with previous co-exposure to EGCG, the gene expression analysis of *SpdC* maintained its general patterns. 

Regarding EGCG epigenetic modulation potential, here we assessed *orfX* transcriptional effects, one of the most relevant epigenetic modulator genes in *Staphylococcus* [17]. *OrfX* is conserved among all staphylococci, being constitutively produced during growth [17,48,49]. The *OrfX* gene product has been suspected of playing an important role in bacterial growth and survival since it is present in every sequenced coagulase-positive or coagulase-negative staphylococcal genome [17,49,50]. *OrfX* methylates 70 S ribosomes, constituting a staphylococcal ribosomal methyltransferase of *RlmH* type [49]. Previous studies have reported that, in bacteria, ribosome methylation provides either moderate resistance to antibiotics or, on the contrary, determines susceptibility to antibiotics, thus affecting bacterial adaptation and resistance [17,49,50]. Here, we observed that *OrfX*, particularly in the MRSA strains, revealed an increased expression of staphylococcal ribosomal methyltransferase across all strains, reaching a maximum value of 2.97 ± 0.35-fold change. These results suggest that *OrfX*-mediated ribosomal methylation is affected by EGCG exposure, playing an essential role in determining phenotype resistance reversion as an epigenetic modulator. 

In recently published studies, with a microarray analysis performed on *S. aureus* treated with or without 500 µg/mL of EGCG, the authors reported differentially expressed genes [50], which is in agreement with our results. A significant increase in transcriptional levels was observed in the genes associated with membrane transport, while decreased transcription was observed in the genes involved in toxin production and stress responses [50]. In the same study, by measuring the membrane potential of the cells treated with or without EGCG, the authors also concluded that EGCG markedly decreased membrane potential, indicating damage to the cell membrane [50]. These results support the hypothesis that EGCG can potentially serve as a natural antimicrobial agent, which is also in agreement with our published study regarding MSSA and MRSA commensal strains [51]. Previously, we have also demonstrated that in *S. aureus* strains isolated from commensal flora of healthy volunteers with associated divergent resistant phenotypes, EGCG exposure resulted in altered transcriptional expression patterns of epigenetic modulators, namely *orfx*, *spdC* and *WalKR*, with notably higher significance for the most susceptible strains [51]. 

Despite the fact that information regarding the epigenetic mechanisms associated with antimicrobial resistance is still scarce, recently, Ghosh D et al., 2020 [52] revised the evidence concerning the links between epigenetics and antibiotic resistance, including adaptive resistance. The authors also evidenced phenotypic heterogeneity among bacterial and epigenetic mechanisms that may contribute to resistance development. The transient nature of the epigenetic marks and associated mechanisms makes it a plausible target for new therapeutic approaches. Furthermore, the associations described between conserved DNA methyltransferases and virulence, host colonization and biofilm formation, among others, suggest that DNA MTases should be considered promising targets for the development of new compounds for biomedical applications [53]. 

Overall, our results clearly demonstrate that EGCG exposure can alter the expression patterns of key epigenetic and drug response genes in *S. aureus* with associated divergent resistant profiles and should be further investigated, potentially as a natural antimicrobial agent and or a therapeutic adjuvant. 

## 4. Materials and Methods

### 4.1. Bacterial Strains Identification and Antibiotic-Resistant Profile Assessment

*S. aureus* strains, including MSSA and MRSA (N = 16 and N = 16, respectively), were isolated from infected patients in the Pathology Laboratory of a Lisbon hospital. After identification through automatized methods (VITEK MS e VITEK 2—Biomerieux, Durham, NC, USA) and antibiogram in VITEK2 (AST648 ref # 420,857 Biomerieux, Durham, NC, USA), the strains were isolated for further analysis. The Hospital Ethics Committee previously approved this study, and all of the inherent ethical principles were properly safeguarded.

The isolated strains identified as having a sensitive methicillin resistance phenotype were collected from hospital infections, such as purulent exudates (collected by a swab or aspirated), exudates from operative wounds, expectorations, urine, superficial exudate (swab and aspirate), dialysis liquid, biopsies/tissue fragments and blood culture. Table 3 summarizes the sample collection sites and the sensitivity and resistance to the antibiotics used in the VITEK2—AST P648 chart (ref # 420,857 Biomerieux, Durham, NC, USA). All of the samples were coded in order to maintain the anonymity of each patient. For the antibiotic sensitivity test, the antibiotics used were as follows: penicillin; tetracycline; trimethroprim/sulfamethoxazole; teicoplanin; linezolid; mupirocin; fusidic acid; rifampicin; vancomycin; oxacillin; gentamycin; erythromycin; moxifloxacin; levofloxacin; cefoxitin; daptomycin; and clindamycin. All of the isolated methicillin-susceptible *S. aureus* (MSSA) and MRSA exhibited a sensitive phenotype for tetracycline. In further analysis, only phenotypes resistant to amoxicillin, gentamicin and imipenem were considered.

### 4.2. EGCG Synergism Evaluation

The synergistic potential of EGCG (E4143 Sigma) was analyzed as described elsewhere [51]. Briefly, we prepared an inoculum of the *S. aureus* strains isolated from infected patients in 1 mL of Mueller Hinton Broth with EGCG at final concentrations of 250 µg/mL, 100 µg/mL, 50 µg/mL and 25 µg/mL at 0.5 McFarland turbidity. The bacterial inoculums were immediately plated on agar and incubated for 18 h, 24 h and 48 h at 37 °C. In order to analyze the bacteria’s recovery capacity after EGCG exposure, MSSA and MRSA strains, with previous co-exposure to EGCG, were subcultured for 24 h with no EGCG. Antibiotic susceptibility was assessed using the disk diffusion method seeded in Mueller Hinton Agar with commercial discs of amoxicillin (25 µg), tetracycline (30 µg), gentamicin (30 µg) and imipenem (10 µg). After incubation, the antibiotic disk inhibition zone was measured, and the susceptibility was assessed using EUCAST Clinical Breakpoint Tables v. 10.0, valid from 1 January 2020 [21]. The values of the measurement intervals of the inhibition zones for imipenem and amoxicillin were inferred from the values of cefoxitin, according to the indications in the EUCAST v10.0 table, valid from 1 January 2020. The significant differences between the different times of exposure of the MSSA and MRSA strains were assessed using Student’s *t*-test. *p*-values < 0.02 were considered significant for EGCG exposure per se, and *p*-values < 0.01 were considered significant for EGCG co-exposure with imipenem, tetracycline, gentamycin and amoxicillin.

### 4.3. FTIR-Spectroscopic Analysis

Fourier-Transform Infra-Red (FTIR) spectroscopic analysis was conducted as described elsewhere [26,54]. 

Fourier-Transform Infra-Red (FTIR) spectroscopic analysis was based on a transmission detection mode associated with a high-throughput reading system. Briefly: duplicates of 5 μL of bacteria cell pellets diluted in water were transferred to a 384-well Si plate and then dehydrated for about 2.5 h in a desiccator under a vacuum pump ME-2 (Vaccubrand, Wertheim, Germany). The spectral data were collected using an FTIR-spectrometer (Vertex 70, Bruker) equipped with an HTS-XT (Bruker) accessory. Each spectrum represented 64 coadded scans with a 2 cm^−1^ resolution, which was collected in a transmission mode between 400 and 4000 cm^−1^. The first well of the 384-well plate did not contain a sample, and the corresponding spectra were acquired and used as background, according to the HTS-XT manufacturer. The spectra were pre-processed using atmospheric correction and baseline correction, and max normalization or by second derivative using a Savitzky–Golay filter, with a second-order polynomial over a 15-point window followed by max normalization. In the case of considering second-derivatized spectra, a smaller region of the spectra was always used (between 600 and 1700 cm^−1^ and between 2800 and 3050 cm^−1^) to minimize the effect of the derivatives on signal noise amplification. Atmospheric corrections were conducted with OPUS^®^ software, version 6.5 (Bruker, Bremen, Germany), whereas the remaining pre-processing methods and processing methods, such as principal component analysis (PCA), hierarchical cluster analysis (HCA) and Partial Least Square (PLS) regression models were conducted with The Unscrambler^®^ X, version 10.4 (CAMO Software AS, Oslo, Norway). The PCA was based on the NIPALS algorithm; HCA was based on Spearman’s rank correction (distance measure) and complete linkage (clustering method). 

### 4.4. Quantitative Real-Time PCR Gene Expression Analysis

The strains with identifying established divergences in resistance profiles (MSSA and MRSA) and in synergism with EGCG and imipenem, tetracycline, gentamycin and amoxicillin were selected for transcriptional analysis. 

Quantitative real-time PCR was performed as described elsewhere [51]. The bacteria’s total RNA was extracted from the cell lysates using the NZY Total RNA Isolation kit (Nzytech, Lisbon, Portugal), according to the manufacturer’s protocol. The concentrations of all of the RNA samples were determined by a fluorescence-based assay, the Qubit™ RNA HS Assay Kit in Qubit™ 3.0 Fluorometer (ThermoScientific, Invitrogen, Massachusetts, USA). Then, 1 µg of the total RNA was reverse transcribed to cDNA using random hexamers from the RevertAid RT Kit (ThermoScientific, Invitrogen, Massachusetts, USA ). Quantitative real-time PCR (qRT-PCR) was conducted using a CFX Connect™ Real-Time PCR Detection System (Bio-Rad, Hercules, CA, USA) for the genes *OrfX*, *SpdC* and *WalKR* and the reference gene 16S rRNA, which was used for data normalization. Each reaction was performed in triplicate using the iQ SYBR Green Supermix (Bio-Rad, Hercules, CA, USA) in a final volume of 20 µL. Control PCRs were also performed for all primer combinations without a template. Specific primers were used for *SpdC* (20), *WalKR* (20), *OrfX* (GenBank accession number (National Center for Biotechnology) AAW81344.1; Forward primer GGGCAAAGCGACTTTGTATT; Reverse Primer TGGGAATGTCATTTTGCTGA; Product Length 105 bp) and *16S rRNA* (20) with the following cycling conditions: initial activation of 95 °C for 2 min followed by 40 cycles of 95 °C for 30 s, 55 °C for 30 s, and 72 °C for 40 s. Then, those cycles were followed by the acquisition of a melting curve in order to check for primer–dimer formation and contaminations. Relative quantification was undertaken by normalizing the target genes’ threshold cycles (Ct) with the mean Ct of *16S rRNA*. The transcript levels were analyzed by calculating ΔΔCt (ΔΔCt = ΔCt resistant phenotypes−mean ΔCt most resistant strains (control)).

All of the statistical calculations were performed by using IBM SPSS statistics 22 software. The significant differences between the different groups were analyzed using Student’s *t*-test (comparison for two groups), and *p* < 0.01 was considered statistically significant. The results are presented as mean ± standard deviation.

### 4.5. Ethics Statement

This work is included in two projects from the Instituto Politécnico de Lisboa accepted in Escola Superior de Tecnologia da Saúde ethical council reference: CE-ESTeSL-Nº.18-2019 and CE-ESTeSL-Nº. 20-2020. 

## 5. Conclusions

Overall, this study supports the antimicrobial and synergistic potential of EGCG and demonstrates that divergent resistant phenotypes of *S. aureus* strains are associated with the differential expression of epigenetic and drug-resistance modulator genes as well as divergent molecular profiles. To the best of our knowledge, this is the first study to assess and analyze the transcriptional divergence of a key epigenetic modulator of *S. aureus* as well as the key drug-response modulators genes of *spdC* and *walKR* in strains isolated from infected patients (nosocomial strains), which are a serious treat for the health of both patients and clinical staff. We believe that the assessment of these drug-response modulators’ gene expression patterns is relevant for the field as new tools, such as FTIR-spectroscopic analysis, are used to rapidly identify resistant *S. aureus* strains (MRSA/MSSA). Our data indicate a clear modulator effect induced by EGCG exposure and corroborates the potential of EGCG for antimicrobial and/or therapeutic adjuvant treatment against antibiotic-resistant microorganisms.

## Figures and Tables

**Figure 1 antibiotics-12-00519-f001:**
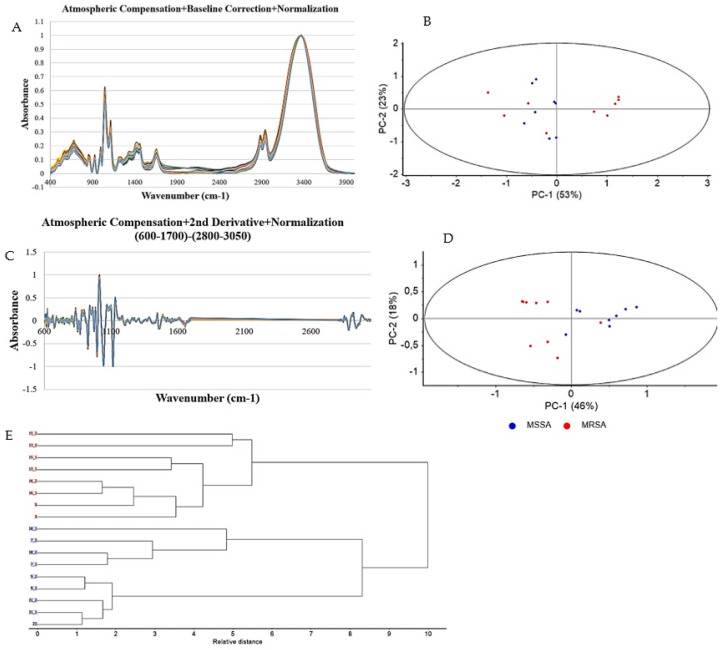
FTIR spectra after baseline and atmospheric correction and normalization (**A**) or normalized second-derivative spectra over (**C**) and its corresponding PCA (**B**,**D**), respectively. The dendrogram from HCA was based on normalized second-derivative spectra (**E**).

**Figure 2 antibiotics-12-00519-f002:**
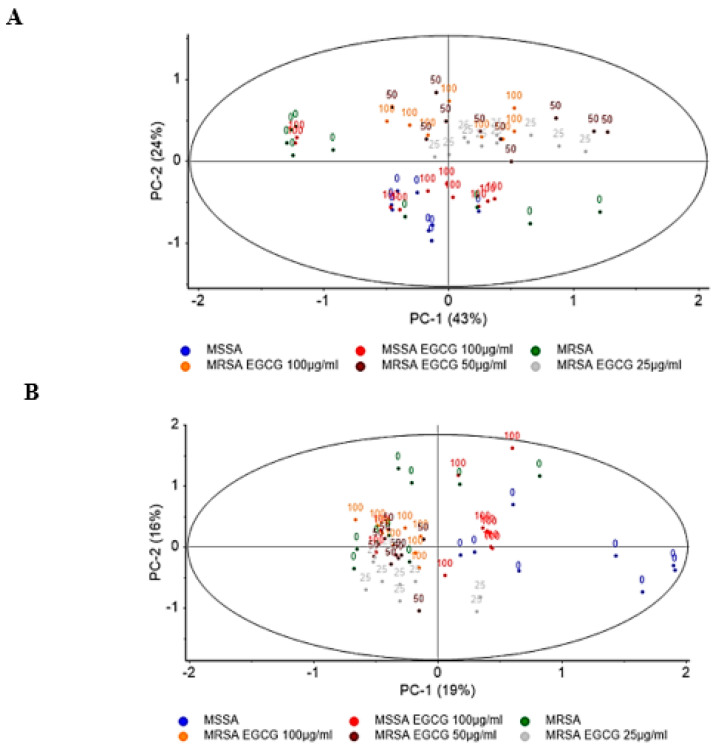
Spectra PCA from MRSA and MSSA incubated with diverse concentrations of EGCG after baseline and atmospheric correction and normalization (**A**) or based on normalized second-derivative spectra (**B**). Colors identify MRSA (green, orange, brown and grey) and MSSA (blue and red), while numbers refer to the EGCG concentration.

**Figure 3 antibiotics-12-00519-f003:**
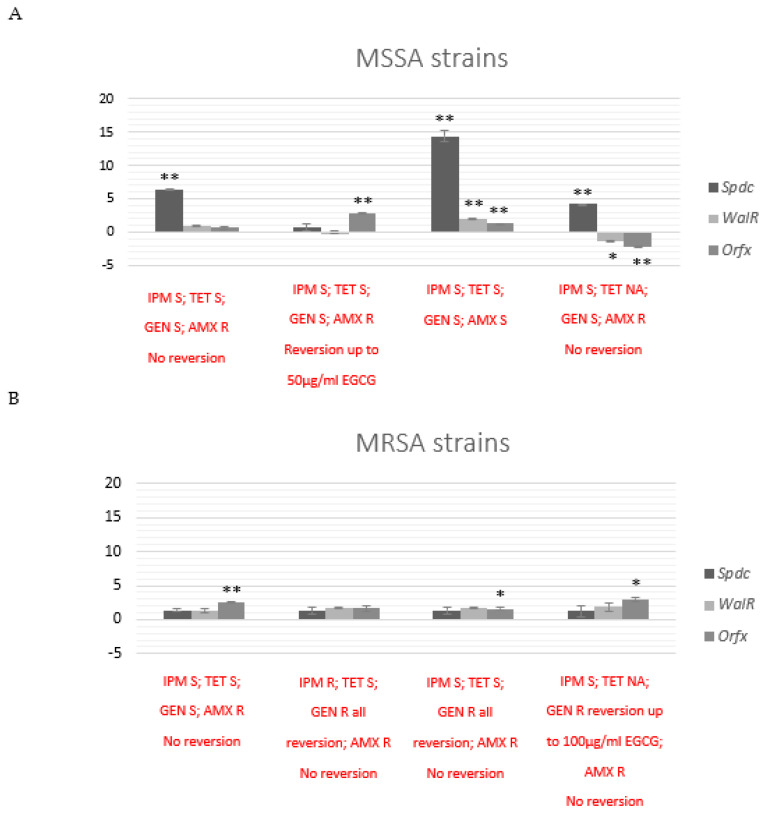
Graphic representation of qRT-PCR analysis of selected MSSA (**A**) and MRSA (**B**) strains after EGCG exposure. Data are representative of the relative expression for the analyzed genes, *OrfX*, *SpdC* and *WalKR*. 16S rRNA was utilized for normalization. Error bars represent the standard deviation between independent treatments and qRT-PCR replicates. We compared the most resistant strains calculated with Student’s *t*-test, for significant statistical values, which were are illustrated as: * *p* < 0.01 and ** *p* < 0.001.

**Figure 4 antibiotics-12-00519-f004:**
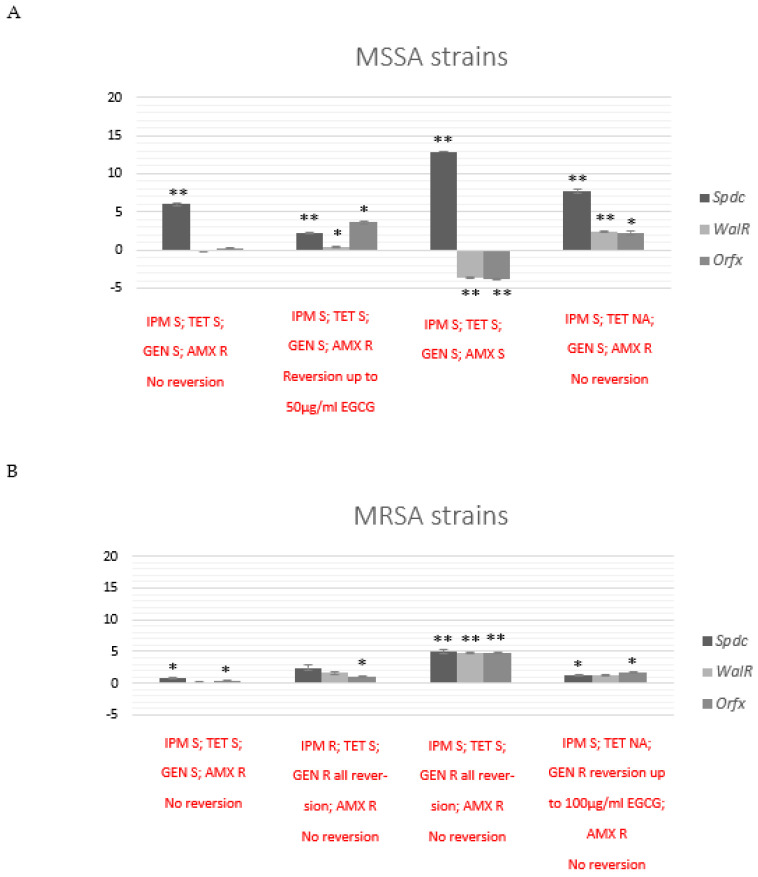
Graphic representation of qRT-PCR data analysis of selected MSSA (**A**) and MRSA (**B**) strains associated with EGCG exposure after 24 h of subculture. Data are representative of *OrfX*, *SpdC* and *WalKR* relative expression. 16S rRNA was used for normalization. Error bars represent the standard deviation between independent treatments and qRT-PCR replicates. Significant statistical values, which were compared with the most resistant strains and calculated with Student’s *t*-test, are illustrated as: * *p* < 0.01 and ** *p* < 0.001.

**Table 1 antibiotics-12-00519-t001:** Resistance profile of isolated MSSA (MS13) and MRSA (MR3, MR4, MR15) strains resistant to gentamycin (30 µg) after co-exposure with EGCG at final concentrations of 250 µg/mL, 100 µg/mL, 50 µg/mL and 25 µg/mL. Antibiotic susceptibility was assessed using the disk diffusion method using EUCAST Clinical Breakpoint Tables v. 10.0, valid from 1 January 2020 [21].

	GEN	GEN + 250 µg/mL EGCG	GEN + 100 µg/mL EGCG	GEN + 50 µg/mL EGCG	GEN + 25 µg/mL EGCG
MS13	R	R	R	R	R
MR3	R	S	S	S	S
MR4	R	S	S	S	S
MR15	R	S	S	R	R

**Table 2 antibiotics-12-00519-t002:** Selected MSSA and MRSA strains for molecular profile analysis and that presented a divergent resistance phenotype. Antibiotic susceptibility was assessed using the disk diffusion method using EUCAST Clinical Breakpoint Tables v. 10.0, valid from 1 January 2020 [21].

Code	Resistance Phenotype
MS3	IPM S; TET S; GEN S; AMX R no reversion
MS5	IPM S; TET S; GEN S; AMX R reversion up to 50 µg/mL EGCG
MS13	IPM S; TET S; GEN S; AMX S
MS15	IPM S; TET NA; GEN S; AMX R no reversion
MS16	IPM S; TET S; GEN R no reversion; AMX R no reversion
MR7	IPM S; TET S; GEN S; AMX All R no reversion
MR9	IPM R no reversion; TET S; GEN R all reversion; AMX all R no reversion
MR18	IPM S; TET S; GEN R all reversion; AMX all R no reversion
MR31	IPM S; TET NA; GEN R reversion up to 100 ugEGCG; AMX all R no reversion
MR22	IPM R no reversion; TET S; GEN S; AMX R no reversion

**Table 3 antibiotics-12-00519-t003:** Antibiotic susceptibility of MSSA (MS) and MRSA (MR) nosocomial isolated strains to OXA (Oxacillin); GEN (Gentamicin); ERY (Erythromycin); CLI (Clindamycin); TET (Tetracycline); SXT (Trimethoprim/Sulfamethoxazole); PEN (Penicillin); TEC (Teicoplanin); MXF (Moxifloxacin); LVX (Levofloxacin); FA (Fusidic Acid); and VAN (Vancomycin).

	Biological Sample	Sensible	Resistent
	MS1: Purulent exudate right foot ulcer	OXA;GEN;ERY;CLI;TET;SXT	PEN;TEC
	MS2: Sputum	OXA;GEN;CLI;LVX;TET;SXT	PEN
	MS3: Purulent exudate left hip	OXA;GEN;ERY;CLI;TET;SXT	PEN;TEC
	MS4: Sputum	OXA;GEN;CLI;LVX;TET;SXT	PEN
	MS5: Blood culture	OXA;GEN;CLI;SXT	
	MS6: Purulent exudate left foot ulcer	OXA;GEN;ERY;CLI;TET;SXT	
M	MS7: Sputum	OXA;GEN;LVX;TET;SXT	ERY;CLI
S	MS8: Superficial exudate aspirated Forearm	OXA;GEN;ERY;CLI;TET;SXT	PEN
S	MS9: Urine	OXA;GEN;CLI;SXT	PEN;MXF;LVX
A	MS10: Superficial exudate aspirated Forearm	PEN;OXA;GEN;CLI;TET; SXT	
	MS11: Sputum	OXA;GEN;CLI;SXT	PEN; ERY;TEC
	MS12: Sputum	OXA;GEN;ERY;CLI;LVX;TET;SXT	PEN
	MS13: Sputum	OXA;CLI;LVX;TET;SXT	GEN; ERY
	MS14: Dialysis Fluid	OXA;GEN;ERY;CLI;TET; SXT	PEN; FA
	MS15: Urine	OXA;GEN;SXT	PEN
	MS16: Exudate surgical wound- Lumbar	OXA;GEN;CLI;TET;SXT	PEN
	MR1: Purulent exudate from wound on back of hand	GEN; SXT;VAN	PEN;OXA;ERY;CLI;MXF;LVX
	MR2: Purulent exudate from right foot ulcer	GEN;ERY;CLI;TET;SXT;VAN	PEN; OXA; FA
	MR3: Sputum	TET; LVX; VAN	PEN; OXA; GEN; ERY;CLI;SXT
	MR4: Bronchoalveolar lavage	TET; VAN	PEN; OXA; GEN; ERY;CLI;SXT
	MR5: Blood culture	GEN;CLI;VAN	PEN;OXA
M	MR6: Purulent swab exudate—schematic foot	GEN;TET;VAN;SXT	PEN;OXA;ERY;CLIMXFLVX
R	MR7: Pressure ulcer biopsy	GEN;TET;SXTVAN	PEN;OXA;ERY;CLI;MXFLVX
S	MR8: Purulent exudate swab—leg	GEN;TET;SXTVAN	PEN;OXA;ERY;CLI;MXF;LVX; FA
A	MR9: Blood culture	GEN;VAN	PEN;OXA;ERY;CLI;MXF;LVX
	MR10: Biopsy/tissue fragment—amputation	GEN;TET;SXT;VAN	PEN;OXA;ERY;CLI;MXF;LVX
	MR11: Blood culture—catéter	GEN;CLI;VAN	PEN;OXA;MXF;LVX
	MR12: Aspirated purulent exudate—inguino-scrotal hernia	GEN; TET; SXT; VAN	PEN;OXA;ERY;CLI;MXF;LVX
	MR13: Purulent swab exudate—burnt skin	GEN;CLI;TET;SXT; VAN	PEN;OXA; FA
	MR14: Exudate Operative wound partial swab-toe matricectomy	GEN;CLI;TET;VAN	PEN;OXA
	MR15: Blood culture	VAN	PEN;OXA;GEN;ERY;CLI; MXF;LVX
	MR16: Blood culture	GEN;VAN	PEN;OXA;ERY;CLI;MXF; LVX

## Data Availability

All data and material will be available as requested.

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
