# Peer review of "Epigenetic and Drug Response Modulation of Epigalocaten-In-3-Gallate in Staphylococcus aureus with Divergent Resistance Phenotypes"

_antibiotics, 2023, doi:10.3390/antibiotics12030519_

Round 1
Reviewer 1 Report
Take care of the writing style, grammar and syntax, as well as the way of citing each of the paragraphs (on the manuscript some detected errors were marked so that the authors attend)
Every reference referred in the text must also present in the reference list and vice versa. In the text, citations should be indicated by the reference number in brackets [3].
References should be written according to the style/form (according to the guidelines suggested by the journal), both in each of the paragraphs they support, and at the end of the reference list (using only the required and important data - in several they do missing data).
It is important to consider the guidelines of writing - editing that gives the journal, and as an example the following conditions can be applied at work: Our results showed that the major constituent of R. officinalis EO was found to be 1,8-cineole, which is in agreement with the study of Badreddine et al. [27], having reported a similar value of major compounds of R. officinalis EO, which is 1,8-cineole (34.8%). Another recent study conducted by Moumni et al. [28], showed that Tunisian R. officinalis EOs are characterized with a high percentage of 1,8-cineole (37.6% to 47.2). Regarding the M. communis EO, it was shown that its major compound was found to be the α-Pinene (35.9%) [29], while the study of Cherrat et al. [30] showed the presence

Author Response
Dear reviewer,
Authors are appreciative and acknowledge the reviewer criticism and hereby reply in concordance with the indicated comments in order to clarify and improve the manuscript.
Take care of the writing style, grammar and syntax, as well as the way of citing each of the paragraphs (on the manuscript some detected errors were marked so that the authors attend)
Authors are appreciative of the reviewer criticism and the manuscript has been revised for English spelling and grammar with the Grammarly premium software. Additionally, all the marked errors were addressed.
Every reference referred in the text must also present in the reference list and vice versa. In the text, citations should be indicated by the reference number in brackets [3].
Authors acknowledge the reviewer criticism and have revised the manuscript in order to include reference number in brackets.
References should be written according to the style/form (according to the guidelines suggested by the journal), both in each of the paragraphs they support, and at the end of the reference list (using only the required and important data - in several they do missing data).
It is important to consider the guidelines of writing - editing that gives the journal, and as an example the following conditions can be applied at work: Our results showed that the major constituent of R. officinalis EO was found to be 1,8-cineole, which is in agreement with the study of Badreddine et al. [27], having reported a similar value of major compounds of R. officinalis EO, which is 1,8-cineole (34.8%). Another recent study conducted by Moumni et al. [28], showed that Tunisian R. officinalis EOs are characterized with a high percentage of 1,8-cineole (37.6% to 47.2). Regarding the M. communis EO, it was shown that its major compound was found to be the α-Pinene (35.9%) [29], while the study of Cherrat et al. [30] showed the presence
Authors acknowledge the reviewer's criticism and have revised the manuscript.
Reviewer 2 Report
The manuscript deals with a relevant topic, but it needs deep restructuring before being able to be published. Some points need an extensive English review.
Abstract and keywords:
According to the journal's rules (https://www.mdpi.com/journal/antibiotics/instructions), the abstract must have a maximum of 200 words and must not have headings (background, methods, results and conclusion). Please make changes in accordance with the guidelines.
Scientific names must always be italicized. Correct the abstract and check the error in the complete manuscript.
The introduction of the abstract is too long in relation to the other items, which prevented the authors from better exploring their results. I suggest a readjustment of the abstract so that it reinforces the key points of the research.
Introduction:
Although the introduction provides relevant information about the state of research in the field and the importance of this work, an extensive English language review is required. Please extend this review to the entire manuscript.
The letters in the paragraph between lines 72 and 79 appear larger than the others. Keep the formatting constant throughout the text. This problem can also be observed in other parts of the manuscript, please correct it throughout the work.
Materials and Methods:
In general, all methods need a better description.
In item 4.2, for how long were the isolates exposed to different concentrations of EGCG? Or, was the bacterial inoculum made with EGCG supplementation and immediately plated on agar to carry out the tests? I advise authors to improve the description of the method, in addition to adding the corresponding reference. Also, on line 469 there is the phrase "(positive control)", but there is no description of it.
Results:
Some of the information provided refers to methodologies, such as the source of the isolates and the antibiotics evaluated. Please rearrange the information in the appropriate section.
Discussion:
The discussion is adequate and in line with the results obtained, only an English review is requested.
Conclusion:
What are the main key points and what is the contribution of this research to the area? Show again some of the main results and show the impact of the research.
Author Response
Dear reviewer,
Authors are appreciative and acknowledge the reviewer criticism and hereby reply in concordance with the indicated comments in order to clarify and improve the manuscript.
The manuscript deals with a relevant topic, but it needs deep restructuring before being able to be published. Some points need an extensive English review.
Authors are appreciative of the reviewer criticism and the manuscript has been revised for English spelling and grammar with the Grammarly premium software.
Abstract and keywords:
According to the journal's rules (https://www.mdpi.com/journal/antibiotics/instructions), the abstract must have a maximum of 200 words and must not have headings (background, methods, results and conclusion). Please make changes in accordance with the guidelines.
Authors acknowledge the reviewer's criticism and have revised the manuscript.
Scientific names must always be italicized. Correct the abstract and check the error in the complete manuscript.
Authors acknowledge the reviewer's criticism and have revised the manuscript.
The introduction of the abstract is too long in relation to the other items, which prevented the authors from better exploring their results. I suggest a readjustment of the abstract so that it reinforces the key points of the research.
Authors acknowledge the reviewer's criticism and have revised the manuscript.
Introduction:
Although the introduction provides relevant information about the state of research in the field and the importance of this work, an extensive English language review is required. Please extend this review to the entire manuscript.
Authors are appreciative of the reviewer criticism and the manuscript has been revised for English spelling and grammar with the Grammarly premium software.
The letters in the paragraph between lines 72 and 79 appear larger than the others. Keep the formatting constant throughout the text. This problem can also be observed in other parts of the manuscript, please correct it throughout the work.
Authors acknowledge the reviewer's criticism and have revised the manuscript.
Materials and Methods:
In general, all methods need a better description.
In item 4.2, for how long were the isolates exposed to different concentrations of EGCG? Or, was the bacterial inoculum made with EGCG supplementation and immediately plated on agar to carry out the tests? I advise authors to improve the description of the method, in addition to adding the corresponding reference. Also, on line 469 there is the phrase "(positive control)", but there is no description of it.
Authors are appreciative and acknowledge the reviewer most relevant comment.
In order to access the synergistic potential of EGCG (E4143 Sigma), we prepared an inoculum of S.aureus strains isolated from patients infections in 1ml of Mueller Hinton Broth with EGCG at final concentrations of 250 µg/ml, 100 µg/ml, 50 µg/ml and 25 µg/ml at 0.5 McFarland turbidity. Bacteria inoculums were immediately plated on agar and incubated for 18 h, 24 h and 48 h at 37° C. In order to analyze bacteria recovery capacity after EGCG exposure, MSSA and MRSA strains, with previous co-exposure with EGCG, were subcultured for 24 h with no EGCG.
The manuscript has been revised accordingly.
Results:
Some of the information provided refers to methodologies, such as the source of the isolates and the antibiotics evaluated. Please rearrange the information in the appropriate section.
Authors acknowledge the reviewer's criticism and have revised the manuscript.
Discussion:
The discussion is adequate and in line with the results obtained, only an English review is requested.
Authors acknowledge the reviewer's suggestion and have revised the manuscript.
Conclusion:
What are the main key points and what is the contribution of this research to the area? Show again some of the main results and show the impact of the research.
Authors are appreciative and acknowledge the reviewer most relevant comment and have revised the manuscript accordingly.
In this study, the isolated nosocomial strains were subjected to the action of four antibiotics including Β-lactams such as amoxicillin, and imipenem (belonging to the carbapenem subgroup) which are able to interfere with the bacterial cell wall synthesis leading to cell lysis. However, considering the work published by Roccaro et al. which proved that catechins interact synergistically with tetracycline against S. aureus here we also included tetracycline and gentamicin, classified as protein synthesis inhibitor antibiotics, in order to access a different mode of action and potential divergences.
To the best of our knowledge this is the first study to assess and analyzed transcriptional divergence of a key epigenetic modulator of S. aureus as well as key drug response modulators genes spdC and walKR. Epigenetic mechanisms, including methylation, are key factors for gene expression and adaptation. In this study we assessed the variation of the OrfX gene expression which methylate 70 S ribosomes, constituting a staphylococcal ribosomal methyltransferase of RlmH type, in relation to differential resistance phenotypes. Additionally, as described in the discussion, drug response modulation has been associated with several factors including WalKR, a two-component system, crucial for S. aureus rapid adaptation to environmental conditions and spdC has been recently classified as new virulence factor for S.aureus. We believe that the assessment of these drug response modulators genes expression patterns is relevant for the field.
Despite of the fact that information regarding epigenetic mechanisms associated with antimicrobial resistance is still scarce, recently Ghosh D el al 2020 (https://doi: 10.1128/AAC.02225-19) summarized evidences of links between epigenetics and antibiotic resistance including adaptive resistance, phenotypic heterogeneity among bacteria’s and epigenetic mechanisms that may contribute to resistance development. The transient nature of epigenetic marks and associated mechanisms makes a plausible target for new therapeutic approaches. Furthermore, Oliveira PH and Fang G 2021 (https://doi: 10.1016/j.tim.2020.04.007.) have also described associations between conserved DNA Methyltransferases and virulence, host colonization, biofilm formation, among others and suggest that DNA MTases should be considered as promising targets for the development of new compounds for biomedical applications.
Additionally, we have also utilized FTIR-spectroscopic analysis to rapidly identify and differentiate resistant S. aureus strains (MRSA/MSSA) and to access divergences in the bacteria molecular profile associated with EGCG exposure.
Reviewer 3 Report
· Pg 1, Line 19-22 ending in …..key modulators. Suggest that these are made two sentences for easy understanding of your aims.
· Pg 1, line 23: “patients infections” might read better as “infected patients”
· Pg 1, line 24: trough = through and in all relevant places, line 25, line 27, lines 28, 68, 372, 468 etc
· Pg 2, line 47-51: Second sentence is too long and should be split into two or three different sentences
· Pg 2, line 52-53: “health associated clinical settings”? Or just clinical setting?
· Pg 2, line 65: sentence part starting with….”leading” is not clear
· Line 363: spelling errors- terapeutic, obteined,
· Line 382-383: “Moreover, we observed 382 that the bacteria whole molecular profile was clearly altered after EGCG exposure for all analysed 383 concentrations, for both MRSA and MSSA strains”…….. Does your data support this assertion? For all strains?.
· Line 546: Reference 2, Geneva
· References are not consistent and require editing
· English language editing would help in understanding the significance and scientific addition of this project to science especially in the introduction and results section. Results and graph interpretations can be improved.
· The authors need to also strongly point out the originality/novelty of project design especially regarding reversal of resistance by green tea and how it differs from the work of others on antimicrobial properties of EGCG.
Author Response
Dear reviewer,
Authors are appreciative and acknowledge the reviewer suggestions and most relevant criticism and hereby reply in concordance with the indicated comments in order to clarify and improve the manuscript.
Pg 1, Line 19-22 ending in …..key modulators. Suggest that these are made two sentences for easy understanding of your aims.
- Pg 1, line 23: “patients infections” might read better as “infected patients”
- Pg 1, line 24: trough = through and in all relevant places, line 25, line 27, lines 28, 68, 372, 468 etc
- Pg 2, line 47-51: Second sentence is too long and should be split into two or three different sentences
- Pg 2, line 52-53: “health associated clinical settings”? Or just clinical setting?
- Pg 2, line 65: sentence part starting with….”leading” is not clear
- Line 363: spelling errors- terapeutic, obteined,
Authors are appreciative of the reviewer criticism and the manuscript has been revised.
- Line 382-383: “Moreover, we observed 382 that the bacteria whole molecular profile was clearly altered after EGCG exposure for all analysed 383 concentrations, for both MRSA and MSSA strains”…….. Does your data support this assertion? For all strains?.
Authors response: FTIR spectroscopy reflects vibrations between atoms within molecules according to the atoms and bonds associated and from interactions due to the atoms surrounding them, in a highly sensitive and specific mode. Consequently, the technique captures the bacteria whole molecular fingerprint, that can be used to discriminate bacteria at the genus, species, and clonal levels (https://doi.org/10.2174/0929867322666150311152800, https://doi.org/10.1016/j.talanta.2019.120347, https://link.springer.com/article/10.1007/s11274-019-2788-5). According to this, in the present work the technique was applied to evaluate the discrimination between MRSA and MSSA strains. It was observed that the spectra of MRSA and MSSA presents distinct molecular features, as pointed by the 100% separation between these strains in spectral hierarchical cluster analysis (Fig. 1).
The high sensitivity and specificity of the technique to capture the bacteria molecular fingerprint has also been used to evaluate the impact of drugs on the bacteria metabolism (https://doi: 10.1002/bit.27915, https://doi: 10.1007/s00253-021-11102-7, https://doi: 10.3390/antibiotics9120897). According to this, in the present work, the technique was applied to evaluate the EGCG impact on MSSA and MRSA strains. It was observed in Fig. 2B, that all MSSA not incubated with EGCG (blue scores) are in a different region of the score-plot after EGCG exposure (red scores), and consequently, all MSSA strains presented a different molecular composition after EGCG exposure. In the Fig. 2A, all MRSA strains not incubated with EGCG (green scores), are also on a different region of the score plot in relation to the same strain after EGCG exposure (grey, brown, and orange colours), independently of the EGCG concentration. Therefore, all MRSA strains presented a different molecular composition after EGCG exposure.
Another interesting observation from Fig2A, is that the distance of scores between strains after EGCG exposure, is higher to the MRSA than with MSSA strains, as highlighted in the top graph. This points that EGCG has a higher impact on the MRSA strains than on the MSSA strains.
This more detailed explanation was incorporated in the revised version of the manuscript.
- References are not consistent and require editing
Authors are appreciative of the reviewer criticism and the manuscript has been revised.
- English language editing would help in understanding the significance and scientific addition of this project to science especially in the introduction and results section. Results and graph interpretations can be improved.
Authors are appreciative of the reviewer criticism and the manuscript has been revised for English spelling and grammar with the Grammarly premium software.
- The authors need to also strongly point out the originality/novelty of project design especially regarding reversal of resistance by green tea and how it differs from the work of others on antimicrobial properties of EGCG.
Authors are appreciative and acknowledge the reviewer most relevant comment.
In this study, the isolated nosocomial strains were subjected to the action of four antibiotics including Β-lactams such as amoxicillin, and imipenem (belonging to the carbapenem subgroup) which are able to interfere with the bacterial cell wall synthesis leading to cell lysis. However, considering the work published by Roccaro et al. which proved that catechins interact synergistically with tetracycline against S. aureus here we also included tetracycline and gentamicin, classified as protein synthesis inhibitor antibiotics, in order to access a different mode of action and potential divergences.
To the best of our knowledge this is the first study to assess and analyzed transcriptional divergence of a key epigenetic modulator of S. aureus as well as key drug response modulators genes spdC and walKR in strains isolated from patients’ infections and thus, nosocomial strains which are a serious treat for the health of booth patients and clinical staff. Epigenetic mechanisms, including methylation, are key factors for gene expression and adaptation. In this study we assessed the variation of the OrfX gene expression which methylate 70 S ribosomes, constituting a staphylococcal ribosomal methyltransferase of RlmH type, in relation to differential resistance phenotypes. Additionally, as described in the discussion, drug response modulation has been associated with several factors including WalKR, a two-component system, crucial for S. aureus rapid adaptation to environmental conditions and spdC has been recently classified as new virulence factor for S.aureus. We believe that the assessment of these drug response modulators genes expression patterns is relevant for the field.
Importantly, despite of the fact that information regarding epigenetic mechanisms associated with antimicrobial resistance is still scarce, recently Ghosh D el al 2020 (https://doi: 10.1128/AAC.02225-19) summarized evidences of links between epigenetics and antibiotic resistance including adaptive resistance, phenotypic heterogeneity among bacteria’s and epigenetic mechanisms that may contribute to resistance development. The transient nature of epigenetic marks and associated mechanisms makes a plausible target for new therapeutic approaches. Furthermore, Oliveira PH and Fang G 2021 (https://doi: 10.1016/j.tim.2020.04.007.) have also described associations between conserved DNA Methyltransferases and virulence, host colonization, biofilm formation, among others and suggest that DNA MTases should be considered as promising targets for the development of new compounds for biomedical applications.
Additionally, we have also utilized FTIR-spectroscopic analysis to rapidly identify and differentiate resistant S. aureus strains (MRSA/MSSA) and to access divergences in the bacteria molecular profile associated with EGCG exposure.
Reviewer 4 Report
The paper entitled “Epigenetic and drug response modulation of Epigalocaten-in-3 gallate in Staphylococcus aureus with divergent resistance phenotypes” reported their findings on how EGCG treatment may reverse antibiotic resistance profiles out of MRSA.
Major concern:
I think overall the paper is written poorly, which makes it extremely difficult to read (major rewriting and reorganization are desperately needed to improve its readability). Also, the result section requires much work to accommodate the journal’s format requirements.
Minor concerns:
I only listed a few here
1 Abstract: Detailed description should be excluded (eg, which automatized method was implemented into this study)
2 Line 47 “nucleic acid” should be referred to as "nucleic acid biogenesis or metabolism"
3 line 80: Can you add citation or reference stating that epigenetic modular in bacteria or human pathogens are seen as drug target? I think it's an overstatement.
4 line 102: Please correct the content here using formal abbreviations for antibiotics (https://journals.asm.org/abbreviations-conventions). It is very confusing and inconvenient to read using the terms created by the authors.
5 line 126: how to rationalize the difference here seen in MR15 but not MR3 or MR4?
6 Line 139-141: Where is the data for this statement? Could you link figure or supplemental figure to it?
7 Figure 3: Why setting the Y axis range up to 20, where the biggest difference you see on the samples are within 5-fold? Also does the Y axis represent fold-change over the WT or over 16s rRNA? There’s no clarity in the figure legend.
Author Response
Dear reviewer,
Authors are appreciative and acknowledge the reviewer criticism and hereby reply in concordance with the indicated comments in order to clarify and improve the manuscript.
The paper entitled “Epigenetic and drug response modulation of Epigalocaten-in-3 gallate in Staphylococcus aureus with divergent resistance phenotypes” reported their findings on how EGCG treatment may reverse antibiotic resistance profiles out of MRSA.
Major concern:
I think overall the paper is written poorly, which makes it extremely difficult to read (major rewriting and reorganization are desperately needed to improve its readability). Also, the result section requires much work to accommodate the journal’s format requirements.
Authors are appreciative and acknowledge the reviewer most relevant criticism and have revised the manuscript in order to clarify potential misreading’s.
In this study, the strains were subjected to the action of four antibiotics including Β-lactams such as amoxicillin, and imipenem (belonging to the carbapenem subgroup) which are able to interfere with the bacterial cell wall synthesis leading to cell lysis. However, considering the work published by Roccaro et al. which proved that catechins interact synergistically with tetracycline against S. aureus here we also included tetracycline and gentamicin, classified as protein synthesis inhibitor antibiotics, in order to access a different mode of action and potential divergences.
To the best of our knowledge this is the first study to assess and analyzed transcriptional divergence of a key epigenetic modulator of S. aureus as well as key drug response modulators genes spdC and walKR. Epigenetic mechanisms, including methylation, are key factors for gene expression and adaptation. In this study we assessed the variation of the OrfX gene expression which methylate 70 S ribosomes, constituting a staphylococcal ribosomal methyltransferase of RlmH type, in relation to differential resistance phenotypes. Additionally, as described in the discussion, drug response modulation has been associated with several factors including WalKR, a two-component system, crucial for S. aureus rapid adaptation to environmental conditions and spdC has been recently classified as new virulence factor for S.aureus. We believe that the assessment of these drug response modulators genes expression patterns is relevant for the field.
Additionally, we have also utilized FTIR-spectroscopic analysis to rapidly identify and differentiate resistant S. aureus strains (MRSA/MSSA) and to access divergences in the bacteria molecular profile associated with EGCG exposure.
Minor concerns:
I only listed a few here
1 Abstract: Detailed description should be excluded (eg, which automatized method was implemented into this study)
Authors acknowledge the reviewer's suggestion and have revised the manuscript.
2 Line 47 “nucleic acid” should be referred to as "nucleic acid biogenesis or metabolism"
Authors acknowledge the reviewer's suggestion and have revised the manuscript.
3 line 80: Can you add citation or reference stating that epigenetic modular in bacteria or human pathogens are seen as drug target? I think it's an overstatement.
Authors are appreciative and acknowledge the reviewer most relevant comment.
However, in line 80 we aimed to explain that orfX is one of the most relevant epigenetic modulators in S. aureus and not particularly a target. “…In Staphylococcus one of the most relevant epigenetic modulators is orfX, which encodes for a 70S ribosomal methyltransferase, whose substrate is S-adenosyl-L-methionine and its C 'terminal is inserted in the SCCmec complex, which contains the mecA gene ([17)].”
Nevertheless, despite of the fact that information regarding epigenetic mechanisms associated with antimicrobial resistance is still scarce, recently Ghosh D el al 2020 (https://doi: 10.1128/AAC.02225-19) summarized evidences of links between epigenetics and antibiotic resistance including adaptive resistance, phenotypic heterogeneity among bacteria’s and epigenetic mechanisms that may contribute to resistance development. The transient nature of epigenetic marks and associated mechanisms makes a plausible target for new therapeutic approaches. Furthermore, Oliveira PH and Fang G 2021 (https://doi: 10.1016/j.tim.2020.04.007.) have also described associations between conserved DNA Methyltransferases and virulence, host colonization, biofilm formation, among others and suggest that DNA MTases should be considered as promising targets for the development of new compounds for biomedical applications.
Thus, authors do not consider an overstatement the assessment of the effects on key epigenetic modulator genes, such as orfX, as new targets.
4 line 102: Please correct the content here using formal abbreviations for antibiotics (https://journals.asm.org/abbreviations-conventions). It is very confusing and inconvenient to read using the terms created by the authors.
Authors are appreciative and acknowledge the reviewer most relevant comment and have revised the manuscript accordingly.
5 line 126: how to rationalize the difference here seen in MR15 but not MR3 or MR4?
Authors are grateful for the reviewer comment.
In our results, MRSA strains encoded as MR3, MR4 that initially presented a gentamicin resistance phenotype, after exposure to all tested concentrations of EGCG, the phenotype has been completely reversed. The strain encoded as MR15 that presented reversion to gentamicin resistance, after exposure to 250µg / m and 100µg / ml of EGCG concentrations, whereas of 50 µg / ml and 25 µg / ml of EGCG the phenotype remained resistant. It is important to notice that these strains are from different anatomic places whereas MR15 was isolated from blood culture and MR3 and MR4 were isolated from respiratory samples (Sputum and Bronchoalveolar lavage, respectively). Additionally, the strains presented divergent resistance phenotypes as the most resistant strain isolated was MR15 which was only sensible to Vancomycin and booth MR3 and MR4 were also sensible to tetracycline. Additionally, MR3 was also previously sensible to levofloxacin and MR15 was resistant. Considering the divergent phenotypes before EGCG exposure it was expected divergent effects after EGCG exposure. The mechanisms associated with the divergent phenotypes are not in the scope of this work. Nevertheless, it is an extremely relevant issue to address in future studies in order com sustain and prove EGCG therapeutic potential
6 Line 139-141: Where is the data for this statement? Could you link figure or supplemental figure to it?
Authors appreciate the reviewer comment.
In the statement “No alterations in imipenem resistance were observed associated with EGCG exposure” we declare that no effects were observed in the analyzed strains after EGCG exposure. Thus, in order to decrease misreading’s, we did not include a figure. Nevertheless, we included in the text the information of “data not shown” with the aim to clarify.
“All MSSA strains isolated had a sensitive phenotype regarding imipenem. On the other hand, MRSA strains coded MR3, MR7, MR11 and MR12 with associated resistance maintained the resistance phenotype after exposure to the assayed different concentrations of EGCG (data not shown).”
7 Figure 3: Why setting the Y axis range up to 20, where the biggest difference you see on the samples are within 5-fold? Also does the Y axis represent fold-change over the WT or over 16s rRNA? There’s no clarity in the figure legend.
Authors acknowledge the reviewer's comment.
In fact, the biggest difference observed on the samples of Figure 3 B are within 5-fold. However, we decided to maintain the Y axis range up to 20 in order to facilitate the comparison of transcriptional expression in relation to Figure 3 A.
Here the relative quantification was undertaken by normalizing threshold cycles (Ct) of the target genes with the mean Ct of 16S rRNA. Transcript levels were analyzed by calculating ΔΔCt (ΔΔCt=ΔCt resistant phenotypes−mean ΔCt most resistant strains (control)). All statistical calculations were performed by using IBM SPSS statistics 22 software. The significant differences between different groups were analysed by a Student’s t-test (comparison for two groups) and p < 0.01 was considered statistically significant. Results are presented as mean ± standard deviation.
Round 2
Reviewer 2 Report
After the review, the authors were able to solve the problems pointed out by the reviewers, and the manuscript is now suitable for publication.
I only consider that better English proofreading is needed, if possible by a native speaker, given the existence of minor grammatical errors.
Reviewer 4 Report
The authors have addressed my concerns.